# Chikungunya in Indonesia: Epidemiology and diagnostic challenges

**Mansyur Arif[1], Patricia Tauran[1], Herman Kosasih[2]\*, Ninny Meutia Pelupessy[1], Nurhayana Sennang[1], Risna Halim Mubin[1], Pratiwi Sudarmono[3], Emiliana Tjitra[4], Dewi Murniati[5], Anggraini Alam[6], Muhammad Hussein Gasem[7], Abu Tholib Aman[8], Dewi Lokida[9], Usman Hadi[10], Ketut Tuti Merati Parwati[11], Chuen-Yen Lau[12], Aaron Neal[12], Muhammad Karyana[2,4]**

**1** Faculty of Medicine, Universitas Hasanuddin/Dr. Wahidin Sudirohusodo Hospital, Makassar, Indonesia, **2** *Indonesia Research Partnership on Infectious Disease (INA-RESPOND), Jakarta, Indonesia, **3** Cipto Mangunkusumo Hospital, Faculty of Medicine Universitas Indonesia, Jakarta, Indonesia, **4** National Institute of Health Research and Development (NIHRD), Ministry of Health, Jakarta, Indonesia, **5** Sulianti Saroso Hospital, Jakarta, Indonesia, **6** Hasan Sadikin Hospital–Faculty of Medicine Universitas Padjadjaran, Bandung, Indonesia, **7** Dr. Kariadi Hospital–Diponegoro University, Semarang, Indonesia, **8** Department of Microbiology, Faculty of Medicine, Public Health and Nursing, Universitas Gadjah Mada, Yogyakarta, Indonesia, **9** Tangerang District Hospital, Tangerang, Indonesia, **10** Dr. Soetomo Academic General Hospital–Faculty of Medicine Universitas Airlangga, Surabaya, Indonesia, **11** Medical Faculty, Udayana University and Sanglah General Hospital, Denpasar, Indonesia, **12** National Institute of Allergy and Infectious Diseases (NIAID), National Institutes of Health, Bethesda, Maryland, United States of America

\* hkosasih@ina-respond.net

**Data Availability Statement:** All relevant data are within the manuscript and its Supporting Information files.

## Abstract

### Background

Chikungunya virus (CHIKV) is often overlooked as an etiology of fever in tropical and sub-tropical regions. Lack of diagnostic testing capacity in these areas combined with co-circulation of clinically similar pathogens such as dengue virus (DENV), hinders CHIKV diagnosis. To better address CHIKV in Indonesia, an improved understanding of epidemiology, clinical presentation, and diagnostic approaches is needed.

### Methodology/Principal findings

Acutely hospitalized febrile patients $\geq$1-year-old were enrolled in a multi-site observational cohort study conducted in Indonesia from 2013 to 2016. Demographic and clinical data were collected at enrollment; blood specimens were collected at enrollment, once during days 14 to 28, and three months after enrollment. Plasma samples negative for DENV by serology and/or molecular assays were screened for evidence of acute CHIKV infection (ACI) by serology and molecular assays. To address the co-infection of DENV and CHIKV, DENV cases were selected randomly to be screened for evidence of ACI. ACI was confirmed in 40/1,089 (3.7%) screened subjects, all of whom were DENV negative. All 40 cases initially received other diagnoses, most commonly dengue fever, typhoid fever, and leptospirosis. ACI was found at five of the seven study cities, though evidence of prior CHIKV exposure was observed in 25.2% to 45.9% of subjects across sites. All subjects were assessed during hospitalization as mildly or moderately ill, consistent with the Asian genotype of

**Funding:** This study was conducted by INA-RESPOND, a collaborative research network of NIHRD, Ministry of Health, Indonesia, and US-NIAID, NIH. This project has been funded in whole or in part with Federal funds from the NIAID, NIH, under contract Nos. HHSN261200800001E and HHSN261201500003I. NIAID collaborators contributed to design of the study; collection, analysis, and interpretation of data; and writing of the manuscript. The content of this publication does not necessarily reflect the views or policies of the Department of Health and Human Services, nor does mention of trade names, commercial products, or organizations imply endorsement by the U.S. Government.

**Competing interests:** The authors have declared that no competing interests exist.

CHIKV. Subjects with ACI had clinical presentations that overlapped with other common syndromes, atypical manifestations of disease, or persistent or false-positive IgM against *Salmonella* Typhi. Two of the 40 cases were possibly secondary ACI.

## Conclusions/Significance

CHIKV remains an underdiagnosed acute febrile illness in Indonesia. Public health measures should support development of CHIKV diagnostic capacity. Improved access to point-of-care diagnostic tests and clinical training on presentations of ACI will facilitate appropriate case management such as avoiding unneccessary treatments or antibiotics, early response to control mosquito population and eventually reducing disease transmission.

## Author summary

Chikungunya virus (CHIKV) is often-overlooked as an etiology of fever in tropical and sub-tropical regions. Lack of diagnostic testing capacity in these areas combined with co-circulation of clinically similar pathogens such as dengue virus (DENV), hinders CHIKV diagnosis. The present study characterized the epidemiology, clinical presentation, and diagnostic approaches to better understand the CHIKV in Indonesia. We screened blood samples collected from children and adults with acute fever admitted to 8 hospitals in Indonesia from 2013–2016. Blood samples negative for DENV infection and a subset blood samples from confirmed DENV cases were then screened for acute CHIKV infection (ACI). ACI was confirmed in 40/1,089 (3.7%) screened patients, of whom initially received other diagnoses such as dengue fever, typhoid fever, and leptospirosis. All patients were considered as mildly or moderately ill, consistent with the Asian genotype of CHIKV. Given the unspecific clinical presentations of ACI, public health measures should support development of CHIKV diagnostic capacity. Improved access to point-of-care diagnostic tests will facilitate appropriate case management, such as avoiding unnecessary treatments or antibiotics, early response to control mosquito population, and eventually reducing disease transmission.

## Introduction

Chikungunya virus (CHIKV) is an important but often overlooked cause of fever in tropical and sub-tropical regions. It sometimes co-circulates with dengue virus (DENV), as both pathogens share the *Aedes* mosquito vector [1,2]. Unlike dengue fever, which is widely recognized in Indonesia, CHIKV infection remains significantly underdiagnosed [3]. Overlapping clinical presentations of DENV, CHIKV and other endemic infections [4] as well as lack of CHIKV testing capacity [5] contribute to underdiagnoses of CHIKV.

Previous reports from the Indonesian Ministry of Health (MOH) have identified multiple CHIKV outbreaks [6–8]. After a hiatus of 16 years, chikungunya outbreaks occurred in 24 areas throughout Indonesia from 2001–2003 [8]. In 2009 and 2010, chikungunya outbreaks hit West and Central Indonesia, and cases rose from approximately 3,000 per year to 83,000 and 52,000 cases per year respectively [6]. After 2010, detected cases fell to 3,000 per year. Except during outbreaks, the incidence rate is likely an underestimate since diagnosis is often based solely on clinical presentation [9,10].

Given the endemicity of CHIKV in Indonesia, a better understanding of CHIKV epidemiology, clinical course, and diagnostic approaches is needed. To address this need, we evaluated demographic, clinical, and laboratory data from patients hospitalized with fever as part of a multi-site observational study conducted in Indonesia.

## Materials and Methods

### Study design and population

Subjects were identified from an observational study of febrile illness [11] conducted by the Indonesia Research Partnership on Infectious Diseases (INA-RESPOND) [12] from July 2013 to June 2016. The study recruited patients ≥1-year-old presenting to one of eight hospitals across Indonesia with acute fever and no history of hospitalization in the past three months. Demographic and clinical information were collected at enrollment; biological specimens were collected at enrollment, once 14 to 28 days after enrollment, and three months after enrollment.

### Specimen screening and testing flow

Blood was collected and processed at study sites and then shipped to INA-RESPOND laboratory. All plasma samples were first screened for DENV infection by Dengue IgM capture/IgG indirect assays, NS1 antigen ELISA (Focus Diagnostics, CA, USA) and/or multiplex semi-nested reverse transcriptase PCR [13]. Specimens negative for DENV infection and a subset specimen from confirmed DENV cases were then screened for acute CHIKV infection (ACI). First, convalescent (3-month) plasma samples were screened by CHIKV IgG ELISA (Euroimmun Ag, Lubeck, Germany). If positive, paired acute and convalescent samples were tested simultaneously by CHIKV IgG/IgM ELISA (Euroimmun Ag, Lubeck, Germany), and acute plasma samples were further tested by real-time reverse transcriptase PCR (rRT-PCR) [14]. Acute plasma from subjects without convalescent specimens were tested by CHIKV IgM/IgG ELISA and rRT-PCR.

### CHIKV serology

Serological testing was performed using anti-CHIKV IgG and IgM ELISA systems according to manufacturer's instructions. Each test used 6 μL of plasma in a 1:100 dilution. Results were read by a microtiter plate reader at 450 nm within 30 minutes of test completion. Samples were considered positive if the optical density (OD) ratio (index) between the sample and calibrator control was ≥1.1.

### CHIKV RT-PCR

**Viral RNA Extraction and RT-PCR.** CHIKV RNA was extracted from plasma using the QIAamp Viral RNA Mini Kit (Qiagen, Hilden, Germany) according to manufacturer's instructions. RNA was eluted in 60 μL and stored at -80˚C until use. The rRT-PCR assay used to detect CHIKV has been previously described [3]. Briefly, a segment in the nonstructural gene of CHIKV was amplified using primers CHIKV-6856 (TCACTCCCTGTTGGACTTGA-TAGA) and CHIKV-6981 (TTGACGAACAGAGTTAGGAACATACC) and QuantiTect Probe RT-PCR Kit (Qiagen, Hlden, Germany), and detected with the probe CHIKV-6919 (3'-FAM-AGGTACGCGCTTCAAGTTCGGCG-BHQ1-5'). The 20 μL reaction mixture consists of 1X QuantiTect Probe RT-PCR Master Mix, 1 μM primers, 0.25 μM probe, 0.25 μL RT-Mix, and 4 μL of extracted CHIKV RNA. The positive control was CHIKV RNA extracted from a CHIKV culture. rRT-PCR was run on the Applied Biosystems 7500 Fast Real-Time PCR

System (Thermo Fisher Scientific, MA, USA) with cycling conditions consisting of reverse transcription at 50˚C for 20 minutes, an initial denaturation at 95˚C for 15 minutes, and 40 cycles at 95˚C for 15 seconds and 60˚C for 60 seconds.

**Genomic sequencing and phylogenetic analysis.** Specimens positive by CHIKV rRT-PCR were sequenced at the E1 gene [3]. Viral RNA was first converted into complementary DNA using the SuperScript III First-Strand Synthesis System for RT-PCR (Invitrogen, CA, USA), and amplified at a region covering the E1 gene using the Platinum Taq DNA Polymerase High Fidelity (Invitrogen, CA, USA) and primer pairs 9870F/10710R and 10378F/11359R. Phylogenetic analysis was done using the maximum likelihood method in the Molecular Evolutionary Genetics Analysis version 7 (MEGA 7.0.20) package. The nucleotide substitution model TN93+G best described the sequence evolution. The strength of tree topology was estimated by bootstrap analysis using 1000 replicates. Trees were unrooted.

**Definition for CHIKV diagnosis classification.** Patients were considered to have ACI when CHIKV RNA was detected by rRT-PCR, sero-conversion occurred or four-fold increase in OD ratio (index) of IgM and IgG between paired samples was observed. Possible secondary ACI was inferred when the above criteria was met and IgG was detected in acute specimens. Probable ACI was inferred when only CHIKV IgM was detected in the acute specimen, but increased index could not be confirmed due to lack of convalescent specimen. Probable ACI with possible secondary infection was inferred when RT-PCR was negative, and IgM sero-converted but OD in acute and convalescent specimens was equivalent. Recent CHIKV infection (RCI) was inferred when CHIKV RNA was not detected, but high index ($>3$) of IgM in acute specimens decreased in convalescent specimens. Non-CHIKV infection with persistent IgM was ascribed when CHIKV RNA was negative, but low index IgM and IgG was detected in acute and convalescent plasma.

## Statistical analysis

Data were collected in OpenClinica v.3.1 (OpenClinica, LLC) and analyzed using STATA v.15.1 (StataCorp LLC). Proportions were compared between groups using the chi-square test. The t-test was used to compare means between groups. Correlation between variables was assessed using the Spearman correlation test.

## Ethics

All subjects or their legal guardians gave their written informed consent for inclusion before they participated in the study. The study was conducted in accordance with the Declaration of Helsinki, and the protocol was approved by the IRBs of Faculty of Medicine University of Indonesia/ Dr. Cipto Mangunkusumo Hospital (451/PT02.FK/ETIK/2012), Dr. Soetomo Hospital (192/Panke.KKE/VIII/2012), and the National Institute of Health and Research and Development (NIHRD), Ministry of Health, Indonesia (KE.01.05/EC/407/2012).

## Results

### Epidemiology

1,486 subjects were enrolled in the parent AFIRE study [15]; 1,464 of those subjects had adequate specimens to be included in this study. One CHIKV case was suspected by site clinicians based on a positive rapid IgM test. However, IgM/IgG ELISA or rRT-PCR at the INA-RESPOND laboratory was negative and *Rickettsia typhi* infection was later confirmed. The remaining 1,463 subjects were screened for DENV, of which 468 (32%) were positive [16]. 105 of 468 randomly selected specimens from confirmed DENV cases were then screened for ACI.

Among the DENV-negative subjects, 12 were excluded due to a lack of adequate specimens, and 983 were screened for CHIKV by IgM/IgG ELISA and/or rRT-PCR. Evidence of ACI was found in 40/1,089 (3.7%) subjects, all of whom were DENV-negative. The screening algorithm and specimen flow is shown in (Fig 1).

The distribution of acute and prior CHIKV infections across sites is shown in (Fig 2). ACI was most prevalent in Makassar (7%), followed closely by Semarang (5.2%). Prior CHIKV exposure, determined by IgG serology, was also highest (37.9% and 45.9%, respectively) in these two cities (Table 1). In Jakarta and Denpasar where the prevalence of prior CHIKV exposure was low (25.2% and 28.5%, respectively), ACI was not detected. Previous exposure was significantly correlated with age group (P<0.01). CHIK cases were identified in Indonesia during every study year, though temporal distribution at sites varied (Fig 3).

## Molecular and serological confirmation in each diagnosis group

**Acute CHIKV infections.** Among 40 ACI, IgM/IgG sero-conversion plus presence of CHIKV RNA was observed in 36 subjects and IgM/IgG sero-conversion alone in 4 subjects. In 35/38 (92%) cases who had late convalescent specimens, IgM was still detected at 3 to 3.5 months after hospitalization. In 2 cases, IgG was also detected in acute specimens, suggesting possible secondary ACI. In 35 of the 36 subjects, rRT-PCR cycle threshold values were low (mean 29.4, range 19.5–39.0). In the other case, CHIKV RNA was detected using a gel-based RT-PCR method. All sequences of CHIKV E1 gene from 30 specimens across 5 study sites were closely related to CHIKV previously identified in Indonesia and belonged to the Asian CHIKV genotype (Fig 4).

**Probable acute CHIKV infection.** Two probable ACI cases with only acute specimens were negative by rRT-PCR, but IgM was detected (titers 1:100 and 1:400) without IgG. In two probable ACI with possible secondary infection cases, IgM sero-converted or increased (0.8 to 3, and 2.4 to 3.3) with high index or slightly increased IgG (3.2 to 3.2 and 2.8 to 3.3).

**Recent CHIKV infection.** In one of the 983 non-dengue subjects, CHIKV IgM between acute and week-three specimens decreased from 4.6 to 2.4 while IgG slightly increased from 3.5 to 3.7, consistent with RCI. An RCI in a subject with dengue was also identified as CHIKV IgM index decreased from 5.9 to 4.7, and IgG index increased from 2.4 to 3.1 between acute and week-two specimens.

**Persistent CHIKV IgM.** In the non-dengue group, low index CHIKV IgM in acute and convalescent specimens, with detectable CHIKV IgG was observed in 9 cases. Two of these cases with persistent IgM were later confirmed as Influenza A/H3N2 and Leptospirosis. In the dengue group, CHIKV IgM and IgG were detected in a DENV-3 infected subject who only had an acute specimen. In another case, low CHIKV IgM titers and high CHIKV IgG titers were detected in acute and convalescent specimens. As CHIKV rRT-PCR was negative, the latter two cases were considered dengue with persistent CHIKV IgM.

**Clinical presentations and laboratory examination.** The 40 ACI subjects had a median age of 20 years (range 1–83 years). The distribution of positive cases by gender (0.67: 1.0, female: male) mirrored that of the parent AFIRE study (0.79: 1.0). Subjects presented to hospital 2–4 days after symptom onset and had a median temperature of 39˚C. Nausea, vomiting, headache, arthralgias, and myalgias were the most common presenting symptoms. Convulsions were significantly more common in children, whereas arthralgias was significantly more common in adults (S1 Table). Most patients had normal leukocyte (78%) and platelet (90%) counts at presentation. However, lymphopenia was slightly more frequent than normal lymphocyte counts [19/35 (54.3%) vs 13/35 (37.1%)]. Hematology profiles were not significantly

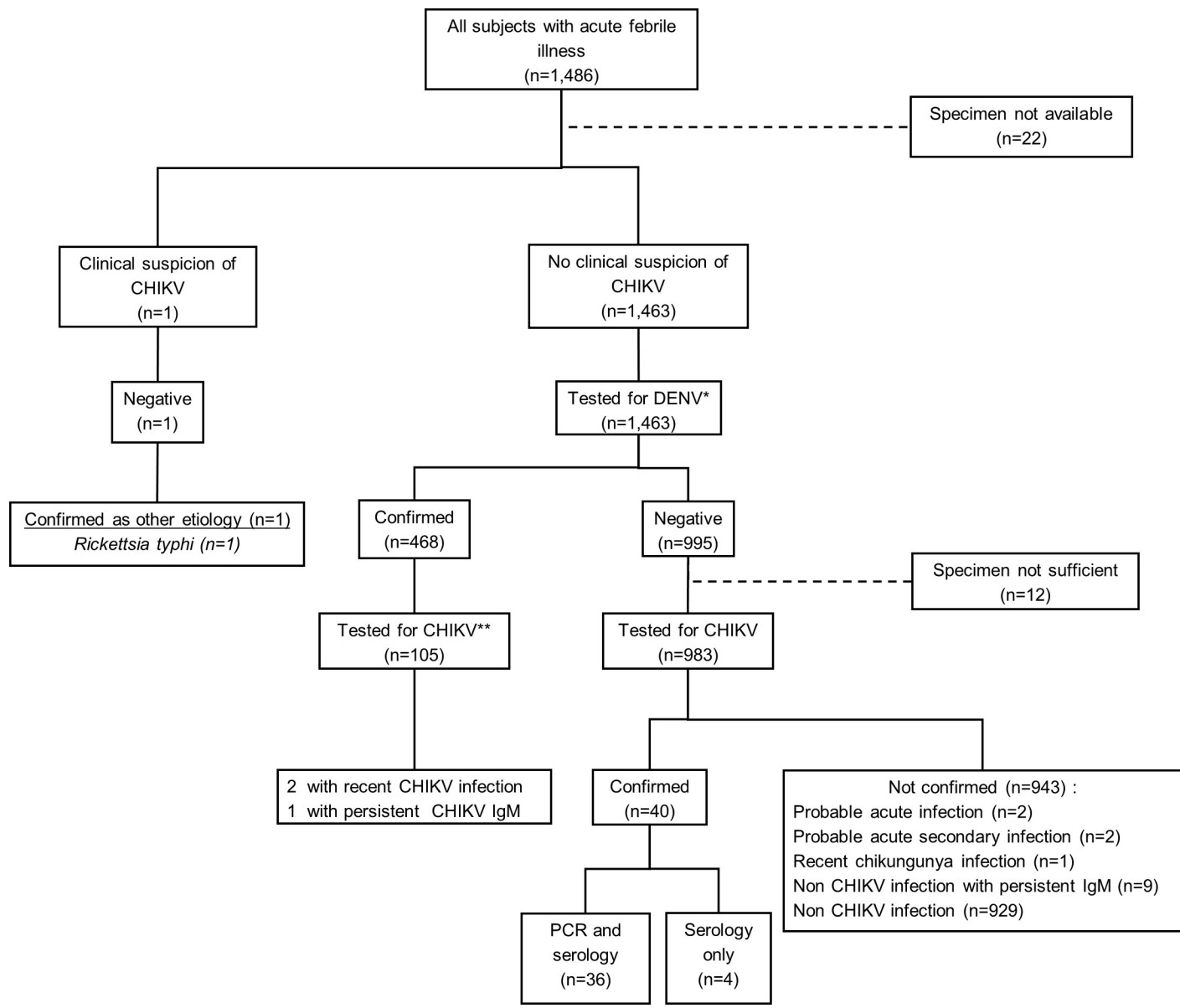

**Fig 1. Study algorithm to identify acute chikungunya infection.** *DENV IgM/IgG, NS1, RT-PCR; ** CHIKV IgM/IgG, RT-PCR.

different between pediatric and adult subjects (S1 Table). Presenting features and clinical laboratory findings are detailed in (Table 2).

None of the 40 ACI cases were accurately diagnosed by study site clinicians. Eleven subjects were misdiagnosed with DENV, two of whom had leukopenia, and all of whom presented with normal platelet counts. All were tested for DENV IgM or DENV NS1 antigen, and only one case was rapid IgM positive, which was later confirmed negative by ELISA. Eight subjects were misdiagnosed with typhoid fever following a positive (score ≥4) S. *typhi* IgM using the TUBEX TF, which was negative by ELISA. One subject was misdiagnosed with leptospirosis despite testing negative for *Leptospira spp* IgM. In these 20 cases, molecular and serological tests for suspected pathogens were all negative.

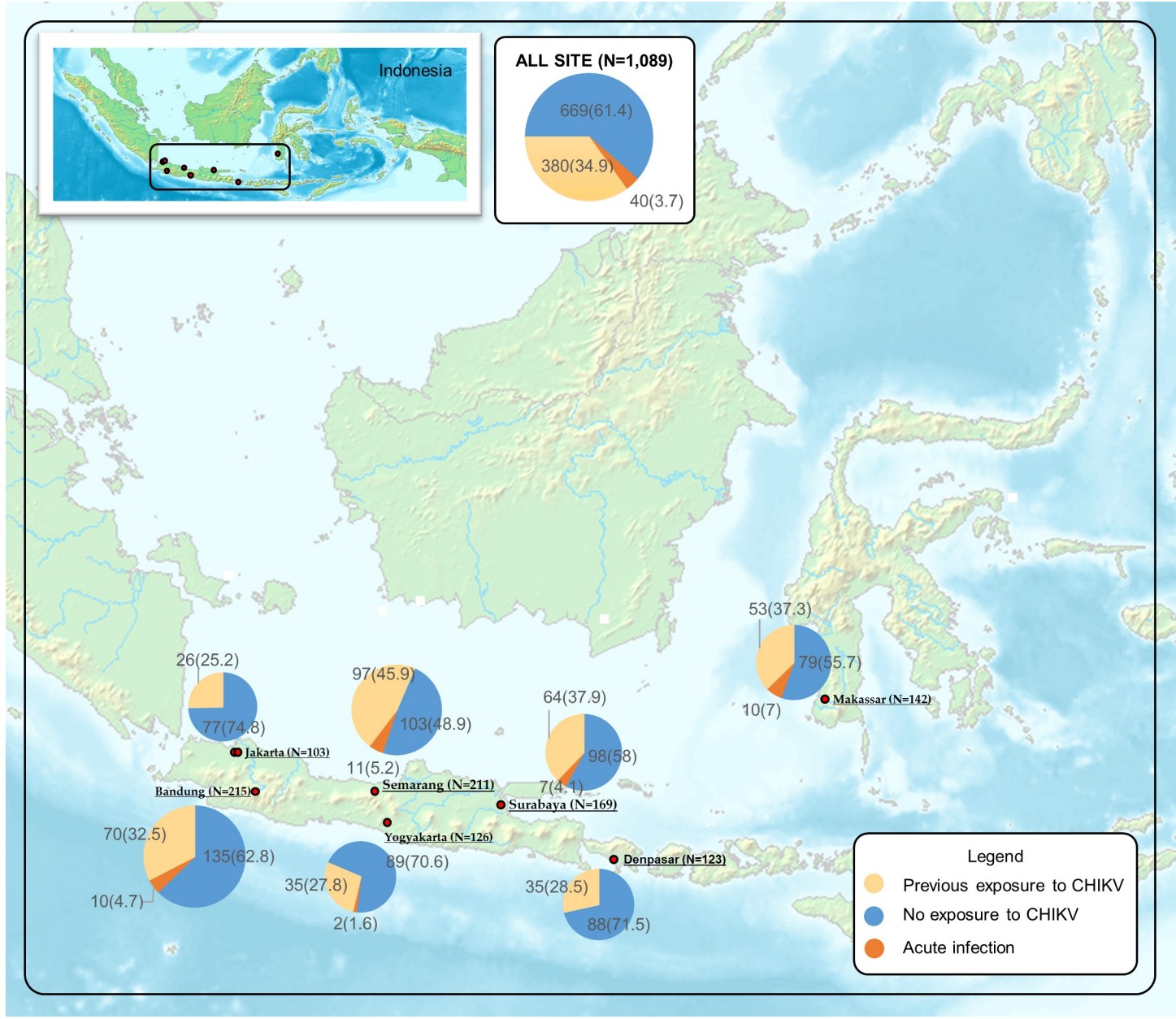

**Fig 2. The distribution of Chikungunya virus exposure in Indonesia.** *Acute CHIKV infection (ACI)*: CHIKV RNA was detected by rRT-PCR and/or sero-conversion or four-fold increase in OD ratio of IgM and IgG between paired samples was observed; *Previous infection*: no evidence of ACI, IgG was positive in acute specimens; *No exposure of CHIKV*: IgG was negative in convalescent specimens. Map source Wikimedia Commons Atlas of the World *[Atlas of Indonesia]*. Available from: https://commons.wikimedia.org/wiki/Atlas_of_Indonesia#/media/File:Map_of_Indonesia_Demis.png [Accessed 21 October 2019].

The remaining 20 subjects were diagnosed with unspecified fever (6), respiratory infection (5), non-specific viral infections (4), fever with rash (2), enteritis (2), and endocarditis (1) (Table 2). In 3 of the 6 unspecified fever cases, dengue or S. *typhi* rapid test was negative. No rapid diagnostic test was done in two elderly patients with suspected lower respiratory infections who had underlying diseases (coronary artery disease and hypertension, and hypertension and osteoarthritis). Rapid dengue or S. *typhi* tests were done in all the 3 cases with upper respiratory tract infection (URTI), and in 3 of 4 with suspected viral infection. All results were

**Table 1. Evidence of prior CHIKV exposure based on IgG sero-positivity by site and age.**

| Site | Age groups Total IgG positive/total tested (%) | | | | | |
|---|---|---|---|---|---|---|
| | 1–5 years | >5–18 years | >18–45 years | >45–65 years | >65 years | All age groups |
| Bandung | 2/34 (5.9) | 20/62 (32.3) | 25/68 (36.8) | 11/30 (36.7) | 12/21 (57.1) | 70/215 (32.5) |
| Denpasar | 0/5 (0) | 1/11 (9.1) | 19/75 (25.3) | 13/28 (46.4) | 2/4 (50.0) | 35/123 (28.5) |
| Jakarta | 3/30 (10) | 6/28 (21.4) | 9/24 (37.5) | 6/14 (42.9) | 2/7 (28.6) | 26/103 (25.2) |
| Makassar | 0/10 (0) | 2/26 (7.7) | 33/79 (41.8) | 15/23 (65.2) | 3/4 (75) | 53/142 (37.3) |
| Semarang | 5/34 (14.7) | 15/52 (28.8) | 38/74 (51.4) | 31/42 (73.8) | 8/9 (88.9) | 97/211 (45.9) |
| Surabaya | 1/14 (7.1) | 6/40 (15) | 29/75 (38.7) | 25/35 (71.4) | 3/5 (60.0) | 64/169 (37.9) |
| Yogyakarta | 6/41 (14.6) | 3/19 (15.8) | 8/28 (28.6) | 9/22 (40.9) | 9/16 (56.3) | 35/126 (27.8) |
| All sites* | 17/168 (10.1) | 53/238 (22.3) | 161/423 (38.1) | 110/194 (56.7) | 39/66 (59.1) | 380/1,089 (34.9) |

*CHIKV exposure is correlated with age group (p<0.01). IgG sero-positivity based on IgG in baseline specimens.

negative. One subject with a cardiac septal defect was suspected of having endocarditis, but blood culture was negative. One of the two diarrhea cases showed decreased consciousness, but no myalgia or arthralgia. Two cases presented with rash, one with cellulitis and severe arthralgia, and the other one was a 2-year-old girl with exanthema subitum. All the clinical and laboratory data are described in detail in (Table 2).

## Discussion

This study reveals important clinical and epidemiologic information about CHIKV in Indonesia. First, CHIKV is an important cause of fever amongst hospitalized patients, but is overlooked by clinicians. Second, prior exposure to CHIKV is high across Indonesian cities and correlates with age. Third, possible secondary ACI may occur. Lastly, all evaluated CHIKV belonged to Asian genotype.

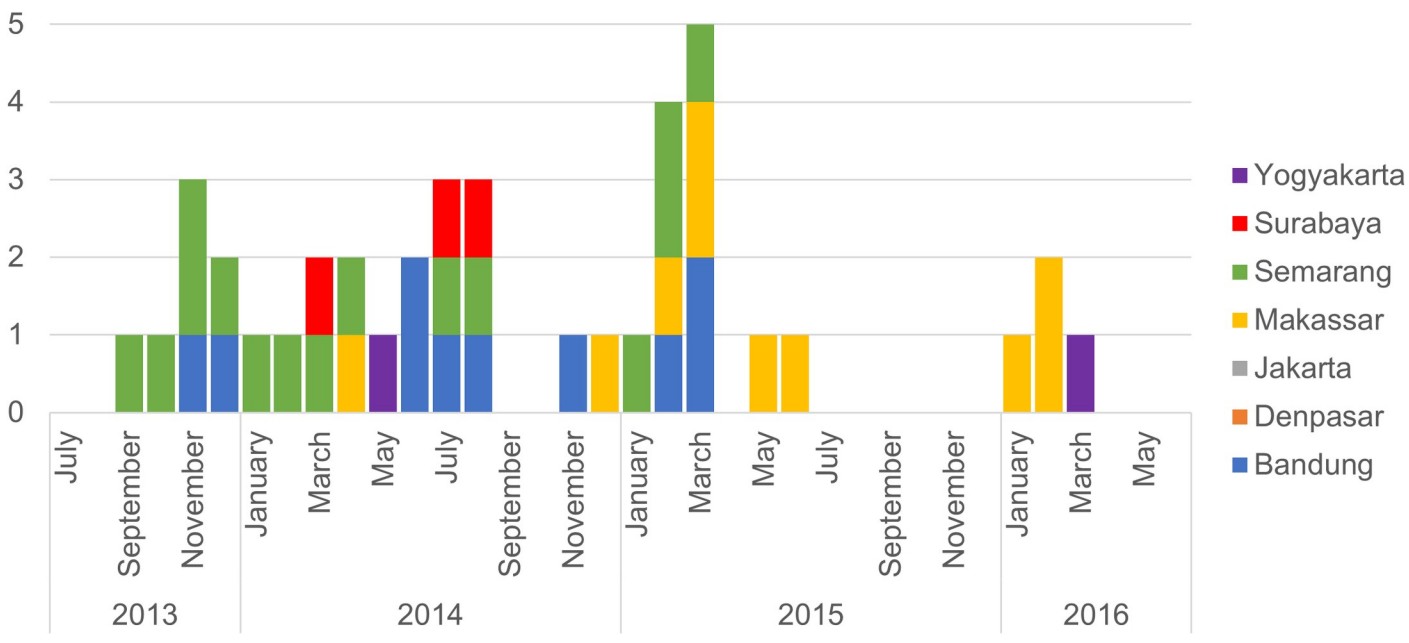

**Fig 3. Distribution of acute chikungunya infection cases by month and year.**

**Table 2. Presenting features of 40 laboratory-confirmed cases of acute CHIKV infection.**

| | Total | Clinical Diagnosis at sites | | | | | | | | |
|---|---|---|---|---|---|---|---|---|---|---|
| | | Dengue (N = 11) | Typhoid fever (N = 8) | Leptospirosis (N = 1) | Fever (N = 6) | Respiratory infection (N = 5) | Viral infection (N = 4) | Endocarditis (N = 1) | Enteritis (N = 2) | Rash (N = 2) |
| **Demographics** | | | | | | | | | | |
| Age, median (range) year | 20.4 (1–83.1) | 15.8 (1.4–71.4) | 22.3 (2.2–39.8) | 37.2 | 3.5 (1–16.3) | 14.4 (4.9–83.1) | 26.3 (23.1–56.1) | 16.1 | 40 (25–56) | 10.8 (1.5–20.1) |
| Female, N (%) | 16 (40) | 4 (36.4) | 4 (50) | 0 | 2 (33.3) | 3 (60) | 0 | 1 | 0 | 2 (100) |
| **Fever** | | | | | | | | | | |
| Days ill, median (IQR) | 2 (1–6) | 2 (1–4) | 2 (1–6) | 6 | 2 (1–4) | 2 (2–5) | 3 (2–5) | 5 | 1.5 (1–2) | 1.5 (1–2) |
| Temperature (°C), median (IQR) | 38.9 (36–41.2) | 39.2 (37.2–39.9) | 38.5 (36–39.8) | 39.5 | 39.1 (38.5–39.8) | 38.7 (38–40) | 38.7 (37.5–39.4) | 38.2 | 40.1 (39.1–41.2) | 37.6 (37–38.2) |
| **Symptoms, N (%)** | | | | | | | | | | |
| **General symptoms** | | | | | | | | | | |
| Anorexia | 10 (25) | 4 (36.4) | 1 (12.5) | 1 | 0 | 2 (40) | 0 | 1 | 0 | 1 (50) |
| Chills | 9 (22.5) | 1 (9.1) | 3 (37.5) | 1 | 2 (33.3) | 0 | 0 | 1 | 0 | 1 (50) |
| Lethargy | 10 (25) | 2 (18.2) | 1 (12.5) | 1 | 1 (16.7) | 1 (20) | 2 (50) | 1 | 1 (50) | 0 |
| Headache | 18 (45) | 5 (45.5) | 4 (50) | 0 | 2 (33.3) | 1 (20) | 4 (100) | 1 | 1 (50) | 0 |
| **Neurological** | | | | | | | | | | |
| Convulsion | 5 (12.5) | 1 (9.1) | 0 | 0 | 4 (66.7) | 0 | 0 | 0 | 0 | 0 |
| Decrease of consciousness | 1 (2.5) | 0 | 0 | 0 | 0 | 0 | 0 | 0 | 1 (50) | 0 |
| **Respiratory** | | | | | | | | | | |
| Cough | 9 (22.5) | 5 (45.5) | 0 | 0 | 2 (33.3) | 2 (40) | 0 | 0 | 0 | 0 |
| Haemoptysis | 1 (2.5) | 1 (9.1) | 0 | 0 | 0 | 0 | 0 | 0 | 0 | 0 |
| Runny nose | 2 (5) | 0 | 0 | 0 | 2 (33.3) | 0 | 0 | 0 | 0 | 0 |
| Epistaxis | 2 (5) | 1 (9.1) | 0 | 0 | 1 (16.7) | 0 | 0 | 0 | 0 | 0 |
| Shortness of Breath | 3 (7.5) | 0 | 0 | 0 | 0 | 2 (40) | 0 | 1 | 0 | 0 |
| **Gastrointestinal** | | | | | | | | | | |
| Diarrhea | 5 (12.5) | 2 (18.2) | 0 | 0 | 0 | 1 (20) | 0 | 0 | 2 (100) | 0 |
| Nausea | 26 (65) | 8 (72.7) | 6 (75) | 1 | 2 (33.3) | 3 (60) | 3 (75) | 1 | 2 (100) | 0 |
| Vomiting | 14 (35) | 5 (45.5) | 2 (25) | 0 | 1 (16.7) | 4 (80) | 1 (25) | 0 | 1 (50) | 0 |

(*Continued*)

**Table 2.** (Continued)

| | Total | Dengue (N = 11) | Typhoid fever (N = 8) | Leptospirosis (N = 1) | Fever (N = 6) | Respiratory infection (N = 5) | Viral infection (N = 4) | Endocarditis (N = 1) | Enteritis (N = 2) | Rash (N = 2) |
|---|---|---|---|---|---|---|---|---|---|---|
| | | | | | **Clinical Diagnosis at sites** | | | | | |
| Epigastric Pain | 5 (12.5) | 2 (18.2) | 1 (12.5) | 0 | 0 | 0 | 2 (50) | 0 | 0 | 0 |
| **Musculoskeletal** | | | | | | | | | | |
| Arthralgia | 18 (45) | 5 (45.5) | 4 (50) | 1 | 0 | 2 (40) | 4 (100) | 1 | 0 | 1 (50) |
| Myalgia | 12 (30) | 4 (36.4) | 2 (25) | 1 | 1 (16.7) | 0 | 3 (75) | 1 | 0 | 0 |
| Skin rash | 6 (15) | 0 | 1 (12.5) | 0 | 1 (16.7) | 1 (20) | 1 (25) | 0 | 0 | 2 (100) |
| Petechiae/ ecchymosis | 4 (10) | 1 (9.1) | 0 | 1 | 1 (16.7) | 1 (20) | 0 | 0 | 0 | 0 |
| **Laboratory findings, N (%)** | | | | | | | | | | |
| Leukocytes* | | | | | | | | | | |
| Leukopenia | 4 (10) | 2 (18.2) | 1 (12.5) | 1 | 0 | 0 | 0 | 0 | 0 | 0 |
| Normal | 31 (77.5) | 8 (72.7) | 6 (75) | 0 | 5 (83.3) | 5 (100) | 4 (100) | 1 | 0 | 2 (100) |
| Leukocytosis | 5 (12.5) | 1 (9.1) | 1 (12.5) | 0 | 1 (16.7) | | 0 | 0 | 2 (100) | 0 |
| Lymphopenia | 19/35 (54.3) | 8 (72.7) | 3/6 (50) | 0 | 3/5 (60) | 3/4 (75) | 0 | 0 | 2 (100) | 0 |
| Normal lymphocytes | 13/35 (37.1) | 3 (27.3) | 3/6 (50) | 0 | 2/5 (40) | 0 | 3/4 (75) | 1 | 0 | 1 (50) |
| Lymphocytosis | 3/35 (7.5) | 0 | 0 | 0 | 0 | 1/4 (25) | 1/4 (25) | 0 | 0 | 1 (50) |
| Platelets | | | | | | | | | | |
| Normal thrombocytes* | 36 (90) | 11 (100) | 6 (75) | 1 | 6 (100) | 5 (100) | 2 (50) | 1 | 2 (100) | 2 (100) |
| Thrombocytopenia (≤150,000/mm³) | 4 (10) | 0 | 2 (25) | 0 | 0 | 0 | 2 (50) | 0 | 0 | 0 |
| **Treatment, N** | | | | | | | | | | |
| Antimicrobial | 21 | 4 | 6 | 1 | 3 | 4 | 0 | 1 | 1 | 1 |
| | | Amoxicillin Chloramphenicol Ceftriaxon Cotrimoxazole | Ceftriaxone (4) Cefixime Amoxicillin | Ceftriaxone | Amoxicillin (2) Ampicillin | Ampicillin Cefixime Ceftriaxone Levofloxacin | - | Cefotaxime | Ciprofloxacin | Ceftriaxone |
| **Outcomes, N (%)** | | | | | | | | | | |
| Cured** | 31 (77.5) | 10 (90.9) | 7 (87.5) | 1 | 5 (83.3) | 4 (80) | 1 (25) | 1 | 2 (100) | 0 |
| Cured with sequelae*** | 9 (22.5) | 1 (9.1) | 1 (12.5) | 0 | 1 (16.7) | 1 (20) | 3 (75) | 0 | 0 | 2 (100) |
| Death before discharge | 0 | 0 | 0 | 0 | 0 | 0 | 0 | 0 | 0 | 0 |

(Continued)

**Table 2.** (Continued)

| | Total | Clinical Diagnosis at sites | | | | | | | | |
|---|---|---|---|---|---|---|---|---|---|---|
| | | Dengue (N = 11) | Typhoid fever (N = 8) | Leptospirosis (N = 1) | Fever (N = 6) | Respiratory infection (N = 5) | Viral infection (N = 4) | Endocarditis (N = 1) | Enteritis (N = 2) | Rash (N = 2) |
| Underlying disease | 6 | 1 Malnutrition | 1 Hypertension | 0 | 0 | 2 CAD & Hypertension (1) Hypertension & Osteoarthritis (1) | 0 | 1 Heart Septal Defect | 1 Lung TB | 0 |
| Diagnostic standard of care test at the hospitals (positive/number tested) | | RDT Den NS1 (0/6) RDT Den IgM (1/7) RDT Salmonella IgM (0/2) RDT Lepto IgM (0/1) | RDT Den NS1 (0/3) RDT Den IgM (0/4) RDT Salmonella IgM (4/8) RDT Lepto IgM (0/1) | RDT Den IgM (0/1) RDT Lepto IgM (0/1) | RDT Den NS1 (0/1) RDT Den IgM (0/1) RDT Salmonella IgM (0/1) | RDT Den NS1 (0/2) RDT Salmonella IgM (0/1) | RDT Den NS1 (0/2) RDT Den IgM (0/1) | Not done | Note done | RDT Den NS1 (0/1) |

* Adjusted by age

**Cured: original illness is no longer present

*** Sequelae: Lethargy, myalgia

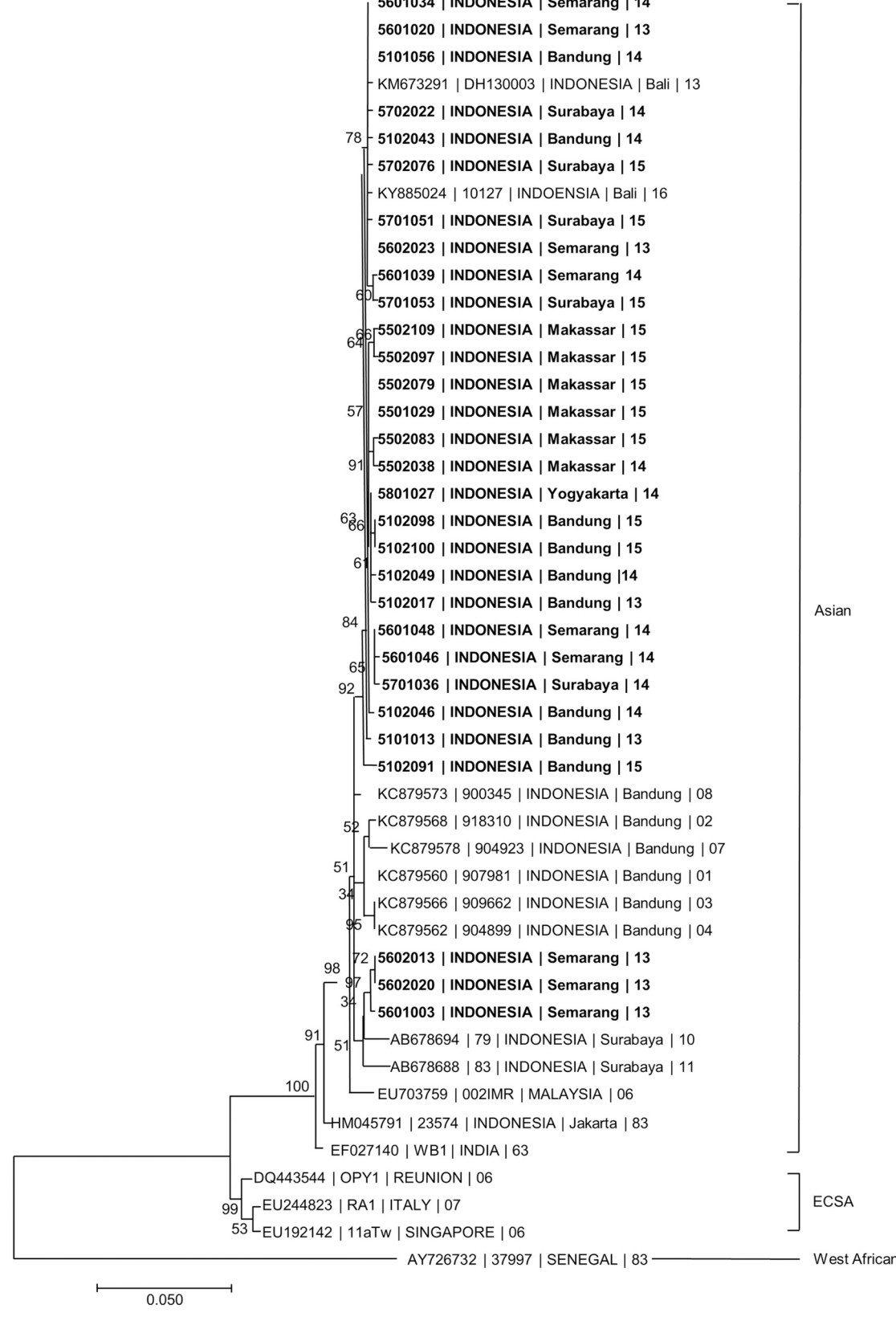

**Fig 4. Phylogenetic trees showing the relationship of Chikungunya virus identified in AFIRE study.** The relationship was constructed by the maximum likelihood method using nucleotide sequences of the E1 gene (1320 bp) with 1000 bootstrap replicates. The trees consist of 30 E1 gene nucleotide sequences from AFIRE specimens and other sequences from GenBank database. The AFIRE specimens are shown in bold, in the following format: Subject Identification Number | Country | City | Year. The sequences retrieved from GenBank are shown in the following format: Accession Number | Strain (if available) | Country | City (if available) | Year. The scale presents the number of nucleotide substitutions per site along the branches. The sequences of the AFIRE Chikungunya E1 gene were submitted to GenBank. The genotypes were inferred based on phylogenetic clustering set by authors referred for the analysis.

CHIKV was the sixth most frequent pathogen identified in the parent AFIRE study, after DENV (32%), *Salmonella spp.* (6.9%), *Rickettsia spp.* (6.9%), influenza virus (4.6%), and leptospirosis (3.3%) [15]. The prevalence of CHIKV in our cohort (2.7%) is similar to that of southern Sri Lanka in 2007 (3%) [17], higher than similar studies in Cambodia (0.6%) [18], Thailand (1.1%) [19] or Central Java (1.7%) [20], but lower than India in 2012 (6%) [21]. Variation in case rates may be affected by different detection methods. The India, Thailand and Central Java studies used ELISA IgM assays in acute specimens only, while Cambodia, Sri Lanka, and our studies used RT-PCR and paired acute and convalescent specimens with ELISA IgM/IgG [17,18]. Evidence of IgM persistence for up to one year [22] may confound the ability to distinguish between acute infection and past exposure, as reflected by the India study

CHIKV infection was not detected in subjects from Jakarta, where evidence of prior exposure was lowest. In Denpasar, which has a high endemic burden of DENV, absence of CHIKV cases in our cohort may be related to competition between these viruses in their vectors [23], or CHIKV was not tested in a sufficient number of dengue confirmed cases (only 21 of 111 (19%) dengue cases from Denpasar, and 105 of 468 (22%) from all sites). However, in Jakarta, it is plausible that *A. aegypti* and *A. albopictus* control programs have reduced the vectors for both CHIKV and DENV. It is also possible that CHIKV and DENV cases were mild and therefore did not present to hospital.

Our study is the first to demonstrate the variation of CHIKV exposure across seven Indonesian cities. CHIKV exposure in our pediatric cohort was much lower than DENV exposure reported or in the AFIRE study [16]; CHIKV and DENV exposure in children ≤5 years and >5–18 years were 10.1% vs 40.2% and 22.3% vs 89%, respectively. This epidemiologic data will benefit public health and immunization programs.

An unexpected result of our study was that none of the 40 laboratory-confirmed cases of CHIKV were recognized by clinicians. This may be attributable to the initial non-specific clinical presentation of the disease, atypical presentations, co-morbidities confounding typical presentations or low clinician awareness.

CHIKV manifestations overlap with those of other endemic infections. Furthermore, CHIKV cases may be subclinical or without pathognomonic manifestations [3, 10]. Arthralgia, the most pathognomonic symptoms for chikungunya, was reported in only 45% of cases. Arthralgia may have been under reported because children may not reliably identify arthralgias and myalgias, as reflected in the lower percentage of children reporting arthralgia compared to adults (23.5% vs 60.9%). Skin rash, which may be difficult to see on patients with dark skin or with minimum light, was even lower (15%). Without the typical signs and symptoms and confounded by presence of non-specific symptoms such as sore throat and cough, clinicians made the diagnosis of undifferentiated fever, febrile convulsions, or URTI in 9 subjects. Also, CHIK may present atypically with diarrhea [24] or decreased consciousness [25]. These were present in 3 of our cases, which were misdiagnosed as enteritis and herpes simplex infection. Comorbidities with chronic diseases such as septal defect, osteoarthritis, or COPD may have led clinicians to suspect typical disease associated infections.

Lack of access to accurate rapid diagnostics, including bedside diagnostics, creates a challenge. Currently CHIKV IgM rapid diagnostics are plagued by low sensitivity. Performance is better for the Eat Central South African (ECSA) genotype, but suboptimal for the Asian genotype circulating in Indonesia [26,27]. And since IgM is usually detectable 5 days after fever onset, utility of IgM in acute specimens is minimal. Additionally, persistence of IgM for CHIKV more than a year after infection [3] may confound interpretation of positive results. Our study reflects these challenges as IgM, even when tested by a sensitive ELISA, was only detected in 10% of CHIK case acute specimens, and also in 11 non-CHIK cases. Furthermore, serological diagnosis requires assessment of titers in convalescent specimens, so has minimal value for acute treatment decisions. These challenges highlight the need for CHIKV RNA or antigen based rapid diagnostic testing to guide clinical decision making.

Availability of reliable point-of-care kits for other pathogens, such as dengue, malaria, or typhoid fever, help rule out those pathogens to narrow the differential. However, even these results should be interpreted cautiously. For example, rapid IgM S. *typhi* may persist [28], which may explain why 8 laboratory-confirmed CHIKV cases were misdiagnosed as typhoid fever in our study. In the absence of accurate rapid diagnostics for CHIK, clinicians will continue to rely on knowledge of local epidemiology, clinical and hematologic data and available diagnostic test results. Exposure history is also helpful as clusters of CHIK are common [29–31]. Headache is often found, but less frequent and severe compared to dengue [32]. Nausea and vomiting was less frequent than dengue, but arthralgia and rash are more frequent [33]. Hematology parameters that can be used to distinguish with dengue are normal leukocyte and normal platelet count, whereas with S. *typhi* were normal leukocyte and lower lymphocyte counts [33].

Differentiating between CHIKV and other infections is important for appropriate case management and anticipation of clinical course [34]. Accurate diagnosis of CHIKV would reduce the unnecessary use of antimicrobials that would be administered if cases were misdiagnosed as typhoid fever, leptospirosis, rickettsiosis or influenza. In our study, more than half of the subjects received unnecessary antibiotics, increasing the risk for adverse events and generation of antibiotic resistance. Accurate diagnosis would also facilitate appropriate patient counseling especially in pregnant patients [35,36].

The identification of two possible secondary ACI in Semarang and Surabaya support previous findings from Bandung [3]. There are several plausible explanations. First, recovery from infection may not engender lifelong immunity. Second, development of CHIK IgG in these patients may have been faster than average. Third, cross-reactivity with other alphaviruses (Ross River virus, Eastern equine encephalitis virus, Barmah-Forest virus, Mayaro virus) [37,38] is possible given that false positives have been reported with the IgG kit we employed when compared to an in-house ELISA assay used by the US-CDC in Panama [39]. However, there were no reports of the circulating Ross river viruses or other alphaviruses in Indonesia, except in Moluccas, which was not a site in our study [38,40]. Furthermore, a study in the Philippines, using the more specific PRNT assay, revealed that CHIK infection only occurred in 106 subjects who did not have CHIKV neutralizing antibodies in their acute samples [41]. Finally, it is also possible that the strains causing the first and second infections were different, which may occur with circulation of both Asian and ECSA genotypes in Indonesia [6]. All CHIKV in our study belonged to the Asian genotype, forming a clade with CHIKV identified in other parts of Indonesia [3,9,42]. Viral genotype has been associated with illness severity, virulence, and transmissibility. The tendency toward milder disease with the Asian genotype compared to the ECSA genotype is consistent with our findings that only one third of patients had sequelae at 3 months and no fatalities occurred [43].

This study has several limitations. As it was part of a larger study to identify etiologies of fever, CHIK specific questions such as those related to arthralgia were not asked. This study was conducted only in large Indonesian cities as opposed to throughout the archipelago, therefore results may not be generalizable to the entire population. Also, CHIKV and DENV co-infections might have been missed; to address this, we evaluated 22% of patients with DENV infection for CHIKV co-infection.

In conclusion, our study highlights the importance of considering CHIKV infection as a cause of acute fever requiring hospitalization in Indonesia. Clinicians must be aware of CHIKV's diverse clinical presentation. Improved recognition and diagnosis will facilitate appropriate treatment and more effective public health measures to mitigate disease transmission and antimicrobial resistance. Development of and improved access to reliable point-of-care diagnostics for CHIKV and other endemic pathogens should be a public health priority. Further studies are needed to confirm co-infection with DENV or other arboviruses and to explore the implications of secondary infection.

## Supporting information

**S1 Checklist. STROBE checklist.**
(DOC)

**S1 Table. Clinical manifestations, hematology profiles, treatment and outcome in pediatrics and adults.**
(DOCX)

**S1 Data. Chikungunya in Indonesia.**
(XLSX)

## Acknowledgments

We would like to thank the following study sites and teams for their participation: Dr. Cipto Mangunkusumo Hospital (Jakarta), Prof. Dr. Sulianti Saroso Infectious Disease Hospital (Jakarta), Dr. Hasan Sadikin Hospital (Bandung), Dr. Kariadi Hospital (Semarang), Dr. Sardjito Hospital (Yogyakarta), Dr. Soetomo Hospital (Surabaya), Sanglah Hospital (Denpasar), and Dr. Wahidin Soedirohusodo Hospital (Makassar). We also would like to thank Aly Diana, Antonius Arditya Pradana, and Nurhayati Lukman for technical assistance of the manuscript.

## Author Contributions

**Conceptualization:** Mansyur Arif, Herman Kosasih, Pratiwi Sudarmono, Emiliana Tjitra, Abu Tholib Aman, Dewi Lokida, Ketut Tuti Merati Parwati, Muhammad Karyana.

**Data curation:** Mansyur Arif, Patricia Tauran, Herman Kosasih, Ninny Meutia Pelupessy, Nurhayana Sennang, Risna Halim Mubin, Pratiwi Sudarmono, Emiliana Tjitra, Dewi Murniati, Anggraini Alam, Muhammad Hussein Gasem, Abu Tholib Aman, Usman Hadi, Chuen-Yen Lau, Aaron Neal, Muhammad Karyana.

**Formal analysis:** Mansyur Arif, Herman Kosasih, Ninny Meutia Pelupessy, Pratiwi Sudarmono, Emiliana Tjitra, Dewi Murniati, Muhammad Hussein Gasem, Abu Tholib Aman, Dewi Lokida, Usman Hadi, Ketut Tuti Merati Parwati, Chuen-Yen Lau, Aaron Neal, Muhammad Karyana.

**Investigation:** Mansyur Arif, Patricia Tauran, Herman Kosasih, Ninny Meutia Pelupessy, Nurhayana Sennang, Risna Halim Mubin, Pratiwi Sudarmono, Emiliana Tjitra, Dewi

Murniati, Anggraini Alam, Muhammad Hussein Gasem, Abu Tholib Aman, Usman Hadi, Chuen-Yen Lau, Aaron Neal, Muhammad Karyana.

**Methodology:** Herman Kosasih, Pratiwi Sudarmono, Emiliana Tjitra, Abu Tholib Aman, Dewi Lokida, Ketut Tuti Merati Parwati, Muhammad Karyana.

**Resources:** Mansyur Arif, Patricia Tauran, Herman Kosasih, Ninny Meutia Pelupessy, Nurhayana Sennang, Risna Halim Mubin, Pratiwi Sudarmono, Emiliana Tjitra, Dewi Murniati, Anggraini Alam, Muhammad Hussein Gasem, Abu Tholib Aman, Usman Hadi, Chuen-Yen Lau, Aaron Neal, Muhammad Karyana.

**Supervision:** Mansyur Arif, Herman Kosasih, Pratiwi Sudarmono, Emiliana Tjitra, Dewi Murniati, Anggraini Alam, Muhammad Hussein Gasem, Abu Tholib Aman, Dewi Lokida, Usman Hadi, Ketut Tuti Merati Parwati, Chuen-Yen Lau, Aaron Neal, Muhammad Karyana.

**Visualization:** Aaron Neal.

**Writing – original draft:** Mansyur Arif, Patricia Tauran, Herman Kosasih, Ninny Meutia Pelupessy, Nurhayana Sennang, Risna Halim Mubin, Chuen-Yen Lau, Aaron Neal.

**Writing – review & editing:** Mansyur Arif, Patricia Tauran, Herman Kosasih, Emiliana Tjitra, Dewi Lokida, Chuen-Yen Lau, Aaron Neal.

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
