## [Decision Letter · Decision Letter 0]

24 Mar 2020

Dear Dr. Kosasih,

Thank you very much for submitting your manuscript "Chikungunya in Indonesia: Epidemiology and diagnostic challenges" for consideration at PLOS Neglected Tropical Diseases. As with all papers reviewed by the journal, your manuscript was reviewed by members of the editorial board and by several independent reviewers. In light of the reviews (below this email), we would like to invite the resubmission of a significantly-revised version that takes into account the reviewers' comments. 

We cannot make any decision about publication until we have seen the revised manuscript and your response to the reviewers' comments. Your revised manuscript is also likely to be sent to reviewers for further evaluation.

Sincerely,

Andrew S. Azman

Deputy Editor

Andrew Azman

Deputy Editor

Reviewer's Responses to Questions

**Key Review Criteria Required for Acceptance?**

**Methods**

-Are the objectives of the study clearly articulated with a clear testable hypothesis stated?

-Is the study design appropriate to address the stated objectives?

-Is the population clearly described and appropriate for the hypothesis being tested?

-Is the sample size sufficient to ensure adequate power to address the hypothesis being tested?

-Were correct statistical analysis used to support conclusions?

-Are there concerns about ethical or regulatory requirements being met?

Reviewer #1: (No Response)

Reviewer #2: -Are the objectives of the study clearly articulated with a clear testable hypothesis stated? Yes

-Is the study design appropriate to address the stated objectives? Yes

-Is the population clearly described and appropriate for the hypothesis being tested? Yes

-Is the sample size sufficient to ensure adequate power to address the hypothesis being tested? Yes

-Were correct statistical analysis used to support conclusions? Yes

-Are there concerns about ethical or regulatory requirements being met? Yes

**Results**

-Does the analysis presented match the analysis plan?

-Are the results clearly and completely presented?

-Are the figures (Tables, Images) of sufficient quality for clarity?

Reviewer #1: (No Response)

Reviewer #2: -Does the analysis presented match the analysis plan? Yes

-Are the results clearly and completely presented? Yes

-Are the figures (Tables, Images) of sufficient quality for clarity? Yes

**Conclusions**

-Are the conclusions supported by the data presented?

-Are the limitations of analysis clearly described?

-Do the authors discuss how these data can be helpful to advance our understanding of the topic under study?

-Is public health relevance addressed?

Reviewer #1: (No Response)

Reviewer #2: - Are the conclusions supported by the data presented? Yes

-Are the limitations of analysis clearly described? Yes

-Do the authors discuss how these data can be helpful to advance our understanding of the topic under study? Yes

-Is public health relevance addressed? Yes

**Editorial and Data Presentation Modifications?**

Reviewer #1: (No Response)

Reviewer #2: (No Response)

**Summary and General Comments**

Reviewer #1: This paper reports on the testing of febrile cases in Indonesia for evidence of CHIKV infection (acute and historic). This paper represents a useful study that quantifies the frequency of CHIKV misdiagnosis as well as the history of CHIKV infection in Indonesia. My only major concern is the over-interpretation of secondary ACI.

I was not convinced by the argument that two cases were probably (or ‘likely’) secondary ACI. This is important in the context of current vaccine development, which relies on immunological markers of lifelong protection. The authors rely on the presence of IgG as measured by an ELISA in an acute sample as evidence of probable secondary acute CHIKV infection. This is at best, weak evidence and should be downplayed. The presence of IgG in an acute sample could be due to cross-reactivity with other alphaviruses or faster than average development of IgG antibodies. I agree that it is an interesting observation but given the substantial uncertainty as to whether they are truly secondary infections, at a minimum I would refer to these as ‘potential/possible secondary ACI’. It should also be discussed in the context of cross-reactivity with other circulating alphaviruses (e.g., see Tesh et al., AJTMH 1975) and a study in the Philippines that showed that all cases of ACI in a Philippines cohort were among those with no detectable antibodies (as measured by more specific PRNTs rather than ELISAs) at baseline (Yoon et al., PLoS NTD). 

The headline figure of 40/1,089 (3.7%) of samples had evidence of acute CHIKV infection is probably not the best figure to use due to the way sampling was conducted – virtually all DENV negative samples were tested however only a subset of DENV positive samples were tested. I would instead report the two figures separately - so XX/XX (XX%) of DENV negative samples had evidence of ACI and XX/XX (XX%) of DENV positive samples have evidence of ACI.

It would be useful to report the month and year of the ACI cases too - maybe even present an epidemic curve.

Reviewer #2: the article is well written and presents interesting results.

PLOS authors have the option to publish the peer review history of their article (what does this mean?). If published, this will include your full peer review and any attached files.

Reviewer #1: No

Reviewer #2: Yes: Luciano Pamplona de Góes Cavalcanti
---

## [Decision Letter · Decision Letter 1]

4 May 2020

Dear Dr. Kosasih,

We are pleased to inform you that your manuscript 'Chikungunya in Indonesia: Epidemiology and diagnostic challenges' has been provisionally accepted for publication in PLOS Neglected Tropical Diseases.

Best regards,

Andrew S. Azman

Deputy Editor

Andrew Azman

Deputy Editor

Reviewer's Responses to Questions

**Key Review Criteria Required for Acceptance?**

**Methods**

-Are the objectives of the study clearly articulated with a clear testable hypothesis stated?

-Is the study design appropriate to address the stated objectives?

-Is the population clearly described and appropriate for the hypothesis being tested?

-Is the sample size sufficient to ensure adequate power to address the hypothesis being tested?

-Were correct statistical analysis used to support conclusions?

-Are there concerns about ethical or regulatory requirements being met?

Reviewer #1: (No Response)

**Results**

-Does the analysis presented match the analysis plan?

-Are the results clearly and completely presented?

-Are the figures (Tables, Images) of sufficient quality for clarity?

Reviewer #1: (No Response)

**Conclusions**

-Are the conclusions supported by the data presented?

-Are the limitations of analysis clearly described?

-Do the authors discuss how these data can be helpful to advance our understanding of the topic under study?

-Is public health relevance addressed?

Reviewer #1: (No Response)

**Editorial and Data Presentation Modifications?**

Reviewer #1: (No Response)

**Summary and General Comments**

Reviewer #1: I have no further comments

PLOS authors have the option to publish the peer review history of their article (what does this mean?). If published, this will include your full peer review and any attached files.

Reviewer #1: No

---

## [Editor Report · Acceptance letter]

26 May 2020

Dear Dr. Kosasih,

We are delighted to inform you that your manuscript, "Chikungunya in Indonesia: Epidemiology and diagnostic challenges," has been formally accepted for publication in PLOS Neglected Tropical Diseases.

Best regards,

Serap Aksoy

Editor-in-Chief

Shaden Kamhawi

Editor-in-Chief
